# RNF138 Downregulates Antiviral Innate Immunity by Inhibiting IRF3 Activation

**DOI:** 10.3390/ijms242216110

**Published:** 2023-11-09

**Authors:** Xianhuang Zeng, Chaozhi Liu, Jinhao Fan, Jiabin Zou, Mingxiong Guo, Guihong Sun

**Affiliations:** 1Taikang Medical School (School of Basic Medical Sciences), Wuhan University, Wuhan 430071, China; xianhuangzeng@whu.edu.cn (X.Z.); 2022203010008@whu.edu.cn (J.Z.); 2Hubei Key Laboratory of Cell Homeostasis, College of Life Sciences, Wuhan University, Wuhan 430072, China; 2022202040105@whu.edu.cn; 3School of Ecology and Environment, Tibet University, Lhasa 850000, China; 2022252040005@whu.edu.cn; 4Hubei Provincial Key Laboratory of Allergy and Immunology, Wuhan 430071, China

**Keywords:** antiviral natural immunity, RNF138, IRF3, PTEN

## Abstract

A viral infection activates the transcription factors IRF3 and NF-κB, which synergistically induces type I interferons (IFNs). Here, we identify the E3 ubiquitin ligase RNF138 as an important negative regulator of virus-triggered IRF3 activation and IFN-β induction. The overexpression of RNF138 inhibited the virus-induced activation of IRF3 and the transcription of the IFNB1 gene, whereas the knockout of RNF138 promoted the virus-induced activation of IRF3 and transcription of the IFNB1 gene. We further found that RNF138 promotes the ubiquitination of PTEN and subsequently inhibits PTEN interactions with IRF3, which is essential for the PTEN-mediated nuclear translocation of IRF3, thereby inhibiting IRF3 import into the nucleus. Our findings suggest that RNF138 negatively regulates virus-triggered signaling by inhibiting the interaction of PTEN with IRF3, and these data provide new insights into the molecular mechanisms of cellular antiviral responses.

## 1. Introduction

The innate immune system is the host’s first line of defense against infection by pathogenic microorganisms [1]. Following viral infection, host pattern recognition receptors (PRRs) recognize pathogen-associated molecular patterns (PAMPs) and activate a series of downstream signaling cascades that induce the production of type I interferons and pro-inflammatory cytokines [1,2,3]. Double-stranded RNA (dsRNA) or single-stranded RNA (ssRNA) generated during a viral infection and replication are classical PAMPs that are detected by various PRRs, including RIG-I-like receptors (RIG-I and MDA5) and Toll-like receptors [4,5]. RIG-I and MDA5 are mainly distributed in the cytoplasm. RIG-I mainly recognizes relatively short dsRNA and cytoplasmic 5′triphosphorylated single-stranded RNA (5′pppssRNA), while MDA5 tends to recognize long-chain dsRNA [6,7]. TLR3, 7, and 8 are mainly distributed in the endosomal membrane, which mainly recognizes viral nucleic acids [8]. TLR3 recognizes dsRNA [9,10], while TLR7 and 8 mainly recognize ssRNA [11,12]. Recent studies report that TLR10 binds dsRNA in vitro at an endosomal pH, indicating that dsRNA is a ligand of TLR10 [13], then activates the adaptor proteins MAVS (also known as IPS-1, VISA, and Cardif), TRAF3, and MyD88, and then leads to the phosphorylation of the downstream kinase TBK1, which promotes the activation of the transcription factors IRF3 and NF-κB and thus induces the transcription and expression of downstream antiviral genes [14,15,16,17,18,19,20,21].

IRF3, the most critical transcription factor in the IFN-β induction pathway, is important in the antiviral innate immune response process [22,23]. Upon recognition of the pathogen by the cell, IRF3 is first phosphorylated and then undergoes a conformational change. IRF3 forms a homodimerization and undergoes nuclear translocation to bind to the promoter sequence of the target gene, the interferon-stimulated response element (ISRE), and induces antiviral gene expression [24,25]. IRF3 activation is finely regulated by post-translational modifications (PTMs) as it undergoes phosphorylation, dimerization, and nuclear translocation [26]. IRF3 is stably and continuously expressed in various cells, and in its resting state, it is located in the cytoplasm as a non-activated monomer. When the virus infects the cell, it causes the C-terminus of IRF3 to be phosphorylated, and its conformation changes to form a homodimer [27,28,29]. In addition, IRF3 acts as a pro-apoptotic factor in the late stage of viral infection, triggering the RLR-induced IRF-3-mediated pathway of apoptosis (RIPA) to eliminate infected cells and invading microorganisms [30]. The important role of IRF3 in the induction of the gene encoding IFN-β suggests that it must be precisely regulated to determine the appropriate immune response to invading viruses [27,31,32,33,34].

The nuclear translocation of IRF3 is tightly controlled by the nuclear export sequence (NES) and NLS [35], which are essential for its activation. Recent studies have reported that USP22 promotes IRF3 nuclear translocation and antiviral responses by deubiquitinating the input protein, KPNA2 [36]. In addition, the Ser97 phosphorylation site of IRF3 is crucial for its entry into the nucleus because our previous results revealed that PTEN exercises a dephosphorylation function on the Ser97 phosphorylation site of IRF3 through its phosphatase activity, thus promoting IRF3 import into the nucleus and activating the expression of type I interferons [37]. However, regulating the nuclear translocation of IRF3 is a complex and dynamic process, and whether other negative regulators are involved in the PTEN-mediated nuclear translocation of IRF3 still requires in-depth study.

RNF138 acts as a ubiquitin-E3 ligase, promotes cell survival by counteracting apoptotic signals as well as directly engaging in apoptosis, and is also involved in DNA damage response, tumourigenesis, neurodegenerative diseases, and chronic inflammatory processes [38,39,40,41,42,43,44]. However, it has been less studied in antiviral innate immunity. It was shown that the African swine fever virus pI215L negatively regulates the cGAS-STING signaling pathway by recruiting RNF138 to inhibit the K63 ubiquitination of TBK1 [45]. Recent studies have shown that the nuclear E3 ubiquitin ligase RNF138, a negative regulator in the inflammatory innate response, represses the LPS-triggered transcription of late inflammatory genes by degrading SMARCC1 of the SWI/SNF complex [46]. However, whether RNF138 inhibits virus-triggered IRF3 activation and IFN-β induction is unclear.

This study found that RNF138 plays an important cytoplasmic role in virus-triggered IRF3 nuclear translocation and cellular antiviral responses. Further studies showed that RNF138 interacts with IRF3 and PTEN. RNF138 inhibits the interaction between PTEN and IRF3 by ubiquitinating PTEN after a viral infection, which inhibits PTEN-mediated IRF3 nuclear translocation, thereby inhibiting virus-triggered IRF3 nuclear translocation and the subsequent expression of downstream genes. Collectively, these findings define a previously unknown function of cytoplasmic RNF138 in antiviral innate immunity and establish a mechanistic link between the triad of RNF138, PTEN, and IRF3 nuclear translocation, thus providing new insights into the molecular mechanisms through which a viral infection triggers IRF3 nuclear translocation.

## 2. Results

### 2.1. RNF138 Negatively Regulates Viral RNA-Triggered Signaling

The E3 ubiquitin ligase RNF138 is a negative regulator in inflammatory responses, suppressing the LPS-triggered transcription of late inflammatory genes through the degradation of SMARCC1 of the SWI/SNF complex [46]. However, whether RNF138 plays a role in antiviral innate immunity is unknown. To investigate whether cytoplasmic RNF138 is also involved in antiviral innate immune signaling, we first explored the effect of RNF138 on SeV (Sendai Virus)-induced reporter gene activation. The reporter assays indicated that an overexpression of RNF138 in HEK293T cells inhibited the SeV-induced activation of the IFNβ, ISRE, and NF-κB promoters (Figure 1A). Quantitative PCR (qPCR) experiments indicated that an overexpression of RNF138 inhibited the SeV-induced transcription of downstream genes, including IFNB1 CXCL10 (Figure 1B). According to the literature, type I interferon induction requires the activation of the transcription factor IRF3 [47]. The overexpression of RNF138 inhibited SeV-induced phosphorylation and the nuclear translocation of IRF3 (Figure 1C,D). These results suggest that RNF138 inhibits virus-triggered IRF3 activation and IFNB1 gene transcription.

### 2.2. Knockout of RNF138 Potentiates Virus-Induced IRF3 Activation and IFNB1 Transcription

Next, to explore whether endogenous RNF138 is required for virus-induced innate immunity signaling (in particular virus-triggered IRF3 activation) under physiological conditions, we generated the RNF138 knockout HEK293T cells by CRISPR-Cas9 and three RNF138 shRNA expression plasmids (#1, #2, and #3), which target different sites in RNF138 mRNA. The immunoblotting results showed that RNF138-KO#1 was more effective in knocking out RNF138 at the protein level (Figure 2A upper panel), which was used for all of the following experiments, whereas shRNF138#1 and #2 were effective in downregulating RNF138 expression at the protein level (Figure 2A bottom panel). Next, we also observed the effect of shRNF138 on SeV-induced ISRE activation. The reporter assays indicated that RNF138 knockdown enhanced SeV-induced ISRE activation in 293T cells compared to the controls (Figure 2B).

To further assess the effect of RNF138 knockout on the expression of the endogenous genes of IFNB1 and ISGs, RNF138 knockout HEK293T cells were infected with SeV. The qPCR experiments indicated that the mRNA levels of IFNB1, CXCL10, ISG15, and IL-6 were significantly higher in the RNF138 knockout HEK293T cells infected with SeV (Figure 2C). Further, we found that RNF138 knockdown significantly enhanced SeV-induced IRF3 phosphorylation, dimerization, and nuclear translocation (Figure 2D–F). These results suggest that RNF138 is a negative regulator of virus-triggered IRF3 activation and IFNβ transcription.

We also generated RNF138 knockout RAW264.7 and L929 cells via the CRISPR-Cas9 method. The qPCR experiments indicated that the mRNA levels of IFNB1 and IL-6 were significantly higher in RNF138 knockout RAW264.7 cells in SeV or VSV (Vesicular Stomatitis Virus) infection (Figure 3A). Consistent with the gene induction assays, SeV- or VSV-induced IRF3 phosphorylation was enhanced in the RNF138 knockout RAW264.7 and L929 cells (Figure 3B,C). Thus, RNF138 is important in virus-triggered IRF3 activation in mouse cells.

### 2.3. RNF138 Negatively Regulates Virus-Triggered Signaling by Targeting IRF3

As mentioned above, the overexpression and endogenous experiments indicate that RNF138 is a negative regulator of innate immune signaling for virus-triggered IRF3 activation. Next, to find the level at which RNF138 regulates the virus-induced activation of the IRF3 activation pathway, HEK293T cells were co-transfected with plasmids expressing RNF138 and the other signal components. RNF138 inhibited ISRE activation mediated by all upstream activators (RIG-1, MAVS, TBK1, IRF3), including the constitutively active phosphorylation mimetic IRF3-5D (Figure 4A). The result suggests that RNF138 functions at the level of IRF3. Then, we checked the interaction between RNF138 and IRF3. The co-immunoprecipitation experiments showed that RNF138 was associated with IRF3 (Figure 4B,C). Importantly, RNF138 binds to IRF3 under physiological conditions, while the interaction between RNF138 and IRF3 is enhanced in the early stage of an SeV infection but weakened in the late stage of the infection (Figure 4D). These results indicated that RNF138 may act as a virus-triggered innate immunity signal at the level of IRF3.

RNF138 contains a domain located at the N-terminal end (RING), a ZNF domain in the middle, and a C-terminal domain (UIM), and IRF3 contains a DBD, TAD transcriptional activation, and RD nuclear response domain [37,38]. To determine which domain of RNF138 interacts with IRF3, the different domain mutants of RNF138 or IRF3 were constructed (Figure 4E,F). The co-immunoprecipitation experiments showed that the RING domain of RNF138 interacts with IRF3 (Figure 4E), whereas the transcriptional activation domain, IAD, of IRF3 interacts with RNF138 (Figure 4F). These results suggest that RNF138 binds to the IAD domain of IRF3 via the RING domain.

RNF138 is an E3 ubiquitin ligase. The aim was to determine whether RNF138 could ubiquitinate IRF3 and target IRF3 for degradation. The reporter assays indicated that the overexpression of RNF138 in HEK293T cells inhibited SeV-induced (left) and IRF3-mediated (right) ISRE activation, but the inhibitory effect was lost in the RNF138 enzyme inactivation mutant (C18A/C54A) (Figure 4G). This result implied that RFN138 depended on its enzymatic activity to participate in the antiviral signaling. Interestingly, we found that the overexpression of RNF138-WT and -CA had little effect on the polyubiquitination of IRF3 (Figure 4H) and the stability of IRF3 (Figure 4I). These results suggested that IRF3 activation regulation by RNF138 may be affected by IRF3-interacting protein ubiquitination.

### 2.4. RNF138 Inhibits IRF3 Activation by Ubiquitinating PTEN

Next, we sought to find regulators that interact with IRF3. Our previous findings revealed that the tumor suppressor PTEN could release the negatively regulated phosphorylation of IRF3 at the Ser97 site upon viral infection, thereby promoting IRF3 nuclear translocation [37]. To determine whether RNF138 interacts with PTEN, the co-immunoprecipitation experiments showed that RNF138 is associated with PTEN (Figure 5A). Interestingly, the overexpression of RNF138 inhibited PTEN interactions with IRF3 (Figure 5B), while the knockout of RNF138 had the opposite effect (Figure 5C). To further determine whether RNF138 affects PTEN ubiquitination, the co-immunoprecipitation experiments showed that the overexpression of RNF138 significantly promoted the polyubiquitination of PTEN (Figure 5D). The qPCR experiments indicated that the PTEN overexpression promoted SeV-induced IFNB1 mRNA expression, which was significantly higher in the RNF138 knockout HEK293T cells (Figure 5E). Furthermore, the RNF138 overexpression inhibited the PTEN-mediated nuclear translocation of IRF3 (Figure 5F). These results suggest that RNF138 may affect PTEN function by promoting PTEN ubiquitination, inhibiting PTEN-IRF3 interactions, and thus inhibiting PTEN-mediated IRF3 nuclear translocation, suppressing IRF3 activation, and antiviral immune responses.

## 3. Discussion

RNF138 is an E3 ubiquitin ligase and is predominantly distributed in the nucleus. At the same time, there is growing evidence that RNF138 protects genomic stability [38,39,40,48] and also plays a role in cancer development [49,50,51]. For example, RNF138 drives NF-κB activation and lymphomagenesis by destabilizing MYD88 L265P through ubiquitination [41]; similarly, it has been shown that RNF138 inhibits colorectal cancer (CRC) by preventing the hyperactivation of NF-κB signaling [51]. However, there are few studies on the role of RNF138 in antiviral innate immunity. The recent literature reports that RNF138 inhibits LPS-triggered late inflammatory gene transcription [46]. However, whether cytoplasmic RNF138 inhibits virus-triggered IRF3 activation and IFN-β induction is unclear.

In this study, we investigated the role of RNF138 in IRF3-mediated signaling. The overexpression of RNF138 significantly inhibited the SeV-induced activation of the ISRE, NF-κB, and IFN-β promoters, as well as SeV-induced phosphorylation, dimerization, and translocation into the nucleus of IRF3, whereas the knockout of RNF138 had the opposite effect. The mouse cell knockout indicates that RNF138 deficiency promotes the SeV- or VSV-induced phosphorylation of IRF3. These findings establish a critical role for RNF138 in the innate immune response to SeV or VSV. Although RNF138 had little effect on IRF3 ubiquitination and did not affect the stability of the IRF3 protein, in combination with reporter assays, the RNF138-CA enzyme inactivation mutant lost the ability to inhibit both SeV-induced and IRF3-mediated ISRE activation, so we suggest that IRF3 activation regulation by RNF138 may be affected by IRF3-interacting protein ubiquitination.

It is well known that the activation of IRF3 prior to its import into the nucleus involves two basic steps: phosphorylation and dimerization. However, our previous results revealed that PTEN exercises a dephosphorylation function on the Ser97 phosphorylation site of IRF3 through its phosphatase activity in the type I interferon-inducible pathway, thereby promoting the nuclear translocation of IRF3 and activating the expression of type I interferon [37]. Therefore, our paper the proposes that the activation of IRF3 and its import into the nucleus may be determined by at least three steps: carboxy-terminal phosphorylation, dimerization, and amino-terminal dephosphorylation. Thus, the positive regulation of IRF3 activation by PTEN-mediated dephosphorylation controls the nuclear translocation of IRF3. In our study, although RNF138 did not affect the ubiquitination of the IRF3 protein, RNF138 affected the ubiquitination of PTEN, which was ubiquitinated and likely inhibited the interaction of IRF3 with PTEN and consequently inhibited the import of IRF3 into the nucleus. We saw that the interactions between IRF3 and PTEN were enhanced after the RNF138 knockout, suggesting that the ubiquitination of PTEN may inhibit the interactions between IRF3 and PTEN. However, the mechanism of the ubiquitination of PTEN by RNF138 and a detailed mechanism for IRF3 activation still need to be further investigated.

We also found that, unlike the knockout of PTEN, which did not affect the phosphorylation and dimerization of IRF3, the knockout of RNF138 promoted the phosphorylation and dimerization of IRF3 (Figure 2D,E), suggesting that RNF138 may also inhibit the phosphorylation and dimerization process of IRF3 by affecting the other IRF3-interacting protein. That is, the RNF138-mediated regulation of innate immunity also has complexity and diversity.

Collectively, these findings define a previously unknown function of cytoplasmic RNF138 in antiviral innate immunity. We confirm the involvement of RNF138 in virus-triggered IRF3 nuclear translocation and how it regulates this process, as well as establish a mechanistic link between the triad of RNF138, PTEN, and IRF3 nuclear translocation, thus providing new insights into the molecular mechanisms by which viral infections trigger IRF3 nuclear translocation (Figure 6). In conclusion, our findings suggest that RNF138 ubiquitinates PTEN and subsequently inhibits PTEN interactions with IRF3, thereby inhibiting the nuclear translocation of IRF3, which provides new insights into the mechanisms that control excessive cellular antiviral responses.

## 4. Materials and Methods

### 4.1. Reagents

This study utilized antibodies against IRF3 (#11312-1-AP), GFP (#50430-2-AP), GAPDH (#10494-1-AP), Lamin B (#12987-1-AP), Actin (#66009-1-Ig), HA (#51064-2-AP), Flag (#20543-1-AP), Myc (#16286-1-AP, all from Proteintch, Wuhan, China), RNF138 (#A10304, ABclonal, Wuhan, China), p-IRF3(Ser386) (#ET1608-22, Huabio, Hangzhou, China), p-IRF3(Ser396) (#29047, CST, Boston, MA, USA), and HRP-conjugated Mouse Anti-Rabbit IgG Light Chain (#AS061, ABclonal, Wuhan, China). Two SYBR qPCR Mixes (ThermoFisher Scientific, Waltham, MA, USA) were purchased from the indicated companies. Sendai virus (SeV) and Vesicular Stomatitis Virus (VSV) were kindly provided by Dr. Hong-Bing Shu of Wuhan University. The TCID50 of SeV is 10^−3.9^/0.1 mL. SeV belongs to the Cantell strain (Charles River Laboratories, Wilmington, MA, USA). The MOI of VSV is 0.1. After virus infection, the HEK293T, RAW264.7, or L929 cells were cultured in an incubator at 37 °C and 5% carbon dioxide for different times. 

### 4.2. Constructs

The cDNAs encoding human RNF138 were amplified and cloned into the pEF vector. The ISRE, IFN-β, NF-κB promoter luciferase reporter plasmids and the mammalian expression plasmids for TBK1, IRF3, IRF3-5D, RIG-Ⅰ, MAVS, HA-Ub and PTEN were kindly provided by Dr. Hong-Bing Shu. Mammalian expression plasmids and prokaryotic expression plasmids for RNF138 were constructed using standard molecular biology techniques.

### 4.3. Transfection and Reporter Gene Assays

The HEK293T cells were seeded in 24-well plates and then transfected with IFN-β-luc, ISRE-luc, or NF-κB-luc, HA-IRF3, shRNF138, RNF138, or its mutant plasmids or an empty control vector via a PEI transfection reagent, together with pRL-TK (Renilla luciferase) as a control. Twenty-four hours (for RNF138 overexpression) or thirty-six hours (for RNF138 knockdown) after transfection, the cells were stimulated with SeV for another 12 h in certain experiments, or the cells were harvested. The cells were lysed with 100 μL passive lysis buffer for 30 min and subjected to measurements of dual-luciferase activity with a dual-specific luciferase assay kit and Luciferase Reporter Assay System (Promega, WI, USA). The firefly luciferase activity was normalized to the Renilla luciferase activity. 

### 4.4. PCR

Total RNA was extracted from cells using TRIzol (Takara, Shiga, Japan). A reverse transcription system (Takara, Shiga, Japan) was used to synthesize cDNA. Two qPCR SYBR Green Master Mixes (Yeasen Biotechnology Co., Ltd., Shanghai, China) and a Bio Rad CFX Connect system were used for qPCR. The mRNA results were normalized to GAPDH expression. The qPCR primer sequences are listed in Appendix A. 

### 4.5. shRNA

The shRNAs targeting RNF138 were constructed by a plasmid pLKO.1-TCR vector and transfected by PEI into HEK293T cells, followed by an immunoblot analysis. The shRNA primer sequences in this study are listed in Appendix A. 

### 4.6. Generation of Knockout Cells by CRISPR/Cas9 Technology

HEK293T, RAW264.7, or L929 cells with the knockout of RNF138 were generated with a Lenti-CRISPR/Cas9-v2 system. HEK293T cells were seeded in 60 mm dishes and transfected with lenti-CRISPR-RNF138-sgRNA#1, sgRNA#2, or lenti-CRISPR-Rnf138-sgRNA, along with the packaging plasmids psPAX2 and pMD2.G via a PEI transfection reagent. The medium was changed 6h after transfection. The supernatants containing lentivirus were harvested 48 h after infection and filtered through a 0.22 μm filter. HEK293T, RAW264.7, or L929 cells were incubated with the lentivirus for 36 h and then selected with puromycin for 7 days. The isolated single-clonal knockout cells were confirmed with western blotting. The sgRNA primer sequences used in this study are listed in Appendix A. 

### 4.7. Co-Immunoprecipitation and Immunoblotting Analyses

For the co-immunoprecipitation experiments, the HEK293T cells were transfected with the indicated plasmid. After 24 h, the cells were harvested and lysed in 0.8 mL of lysis buffer (20 mM Tris, pH 7.5, 150 mM NaCl, 1% TritonX-100, sodium pyrophoshate, β-glycerophosphoate, 1 mM EDTA, Na_3_VO_4_, 10 μg/mL leupeptin). For an immunoprecipitation reaction, 0.7 mL of the cell lysate was incubated with Flag beads or the indicated antibody and 40 μL of protein A/G agarose beads at 4 °C. After overnight incubation, the beads were washed three times with 1 mL of Co-IP washing buffer (50 mM NaCl and 150 mM Tris, pH 7.5). The immunoprecipitates and whole-cell lysate were subjected to SDS-PAGE, transferred onto nitrocellulose membranes, and blotted as described previously [37]. For the endogenous immunoprecipitation experiments, HEK293T cells were seeded in 100 mm dishes and treated with SeV for the indicated times after 16 h. The subsequent procedures were performed as described above. 

### 4.8. Ubiquitination Assays

The HEK293T cells were transfected with the indicated plasmid. After 24 h, the cells were harvested and lysed on ice in 1 mL of lysis buffer, as described above (100 μL, in the presence of 1% SDS). Next, the cell lysates were denatured at 100 °C for 5 min and then diluted to 10 volumes of lysis buffer (without SDS) before incubation with anti-Flag beads at 4 °C for overnight. The beads were washed three times with 1 mL of Co-IP washing buffer and analyzed via immunoblotting. 

### 4.9. Native PAGE

The HEK293T cells were harvested and lysed with an ice-cold lysis buffer (20 mM Tris, pH 7.5, 150 mM NaCl, 1% TritonX-100, sodium pyrophoshate, β-glycerophosphoate, 1 mM EDTA, Na_3_VO_4_, 10 μg/mL leupeptin, without SDS). Next, the cell lysates were diluted with 2× loading buffer (50 mM Tris-HCl, pH 6.8, 30% glycerol, and bromophenol blue, without SDS). Finally, the samples were loaded onto 8% native gels and separated at 20 mA for 30 min and 35 mA for 90 min on ice through electrophoresis, followed by an immunoblot analysis. 

### 4.10. Subcellular Fractionation

Nuclear and cytoplasmic extracts were prepared with a nuclear-cytoplasmic extraction kit (KeRui Biotechnology Co., Ltd., Wuhan, China) according to the manufacturer’s instructions. 

### 4.11. Statistical Analysis

All data were collected for at least two experiments with similar results. The data were processed in Microsoft Excel and Origin 6.0 and are presented as the mean ± SD of one representative experiment. The differences between the control and experimental groups were determined using the Student’s *t*-test. *p* values < 0.05 were considered statistically significant.

## Figures and Tables

**Figure 1 ijms-24-16110-f001:**
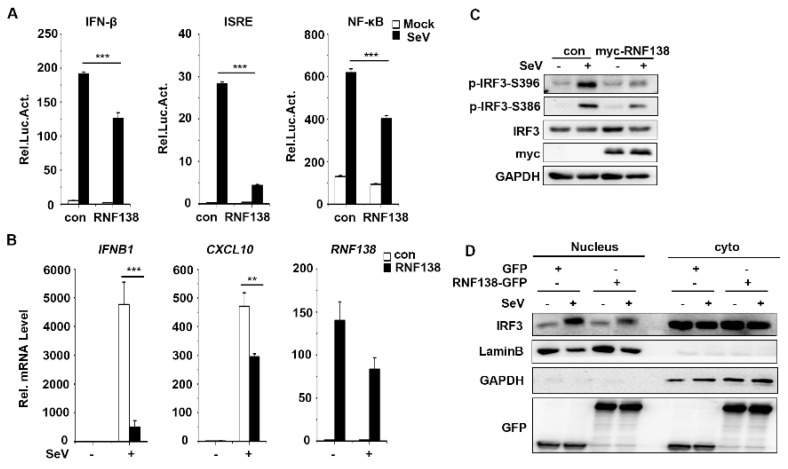
The overexpression of RNF138 inhibited the virus-triggered activations of IRF3 and IFNB1 gene transcription. (**A**) RNF138 inhibited the IFN-β promoter, ISRE, and NF-κB. HEK293T cells were transfected with the IFN-β, ISRE, and NF-κB reporter and the control or RNF138 plasmid for 24 h and then infected with SeV for 12 h before luciferase assays. (**B**) The effects of RNF138 on the SeV-induced transcription of downstream genes. HEK293T cells were transfected with the control or RNF138 plasmid for 24 h and then infected with SeV for 8 h before qPCR analysis. (**C**) Effects of RNF138 on SeV-induced phosphorylation of IRF3 (Ser396, Ser386). HEK293T cells were transfected with the control or myc-RNF138 plasmid for 24 h, then infected with SeV for 8 h before immunoblotting analysis with the indicated antibodies. (**D**) The effects of RNF138 on the SeV-induced nuclear translocation of IRF3. HEK293T cells were transfected with GFP or RNF138-GFP plasmid for 24 h, then infected with SeV for 8 h. Immunoblot analysis of IRF3 in cytoplasmic (Cyto) and nucleus fractions with the indicated antibodies. ** *p* < 0.01, *** *p* < 0.001 (unpaired *t*-test). Data represent at least two experiments with similar results (mean ± SD, n = 3 independent samples in (**A**,**B**,**D**)).

**Figure 2 ijms-24-16110-f002:**
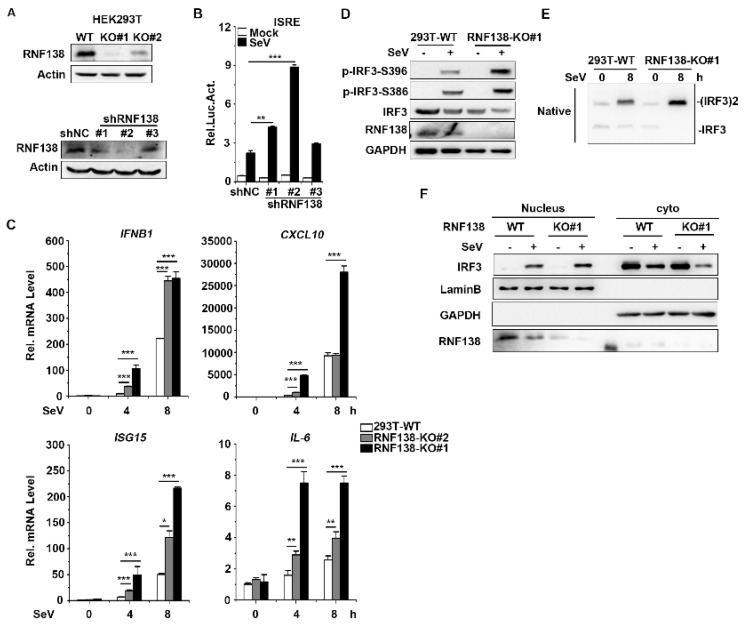
Effects of knockdown and knockout of RNF138 on SeV-induced signaling and IRF3 activation. (**A**) RNF138-deficient (KO) HEK293T clones were generated via the CRISPR-Cas9 method (upper panel) and the effects of RNF138 RNAi on the expression of endogenous RNF138 (lower panel). Deficiency of RNF138 in the KO clones was confirmed through immunoblotting analysis with anti-RNF138. HEK293T cells were transfected with the indicated RNAi plasmids for 48 h before immunoblotting analysis with anti-RNF138. (**B**) Effects of RNF138 knockdown on Sev-induced activation of ISRE. HEK293T cells were transfected with the ISRE reporter and the control or RNF138 RNAi plasmids for 36 h and then infected with SeV for 12 h before luciferase assays. (**C**) The effects of RNF138 deficiency on SeV-induced transcription of downstream genes. RNF138-KO and control HEK293T cells were infected with SeV for the indicated times before qPCR analysis. (**D**) The effects of RNF138 deficiency on SeV-induced phosphorylation of IRF3 (Ser386, Ser396). RNF138-KO and control HEK293T cells were infected with SeV for 8 h before immunoblotting analysis with the indicated antibodies. (**E**) The effects of RNF138 deficiency on SeV-induced dimerization of IRF3. RNF138-KO and control HEK293T cells were infected with SeV for 8 h, and then cell lysates were separated via native PAGE and analyzed by immunoblots with anti-IRF3. (**F**) The effects of RNF138 deficiency on SeV-induced nuclear translocation of IRF3. RNF138-KO and control HEK293T cells were infected with SeV for 8 h, and then immunoblot analysis of IRF3 in cytoplasmic (Cyto) and nucleus fractions was carried out with the indicated antibodies. * *p* < 0.05, ** *p* < 0.01, *** *p* < 0.001 (unpaired *t*-test). Data represent at least two experiments with similar results (mean ± SD, n = 3 independent samples in (**B**–**D**)).

**Figure 3 ijms-24-16110-f003:**
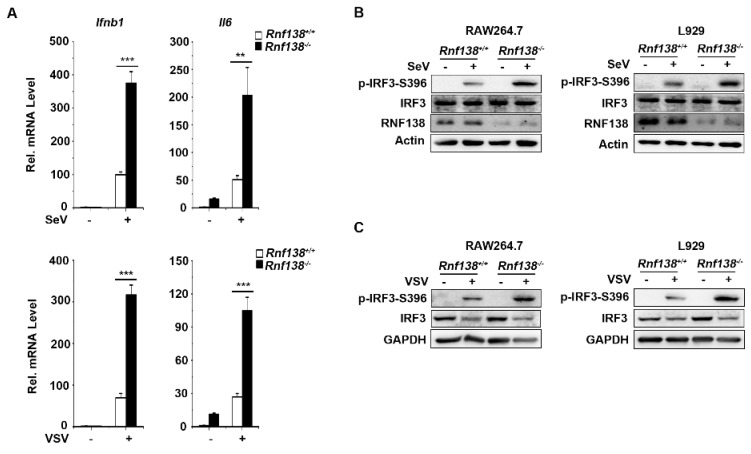
Knockout of RNF138 in mouse cells promotes virus-induced IRF3 activation. (**A**) The effects of RNF138 deficiency on the SeV or VSV-induced transcription of downstream genes. RNF138-KO and control RAW264.7 cells were infected with SeV or VSV for 8 h before qPCR analysis. (**B**) The effects of RNF138 deficiency on SeV-induced phosphorylation of IRF3 (Ser396). RNF138-deficient (KO) RAW264.7 or L929 clones were generated via the CRISPR-Cas9 method. RNF138-KO and the control RAW264.7 or L929 cells were infected with SeV for 8 h before immunoblotting analysis with the indicated antibodies. (**C**) The effects of RNF138 deficiency on the VSV-induced phosphorylation of IRF3 (Ser396). RNF138-KO and control RAW264.7 or L929 cells were infected with VSV for 8 h before immunoblotting analysis with the indicated antibodies. ** *p* < 0.01, *** *p* < 0.001 (unpaired *t*-test). Data represent at least two experiments with similar results (mean ± SD, n = 3 independent samples in (**A**)).

**Figure 4 ijms-24-16110-f004:**
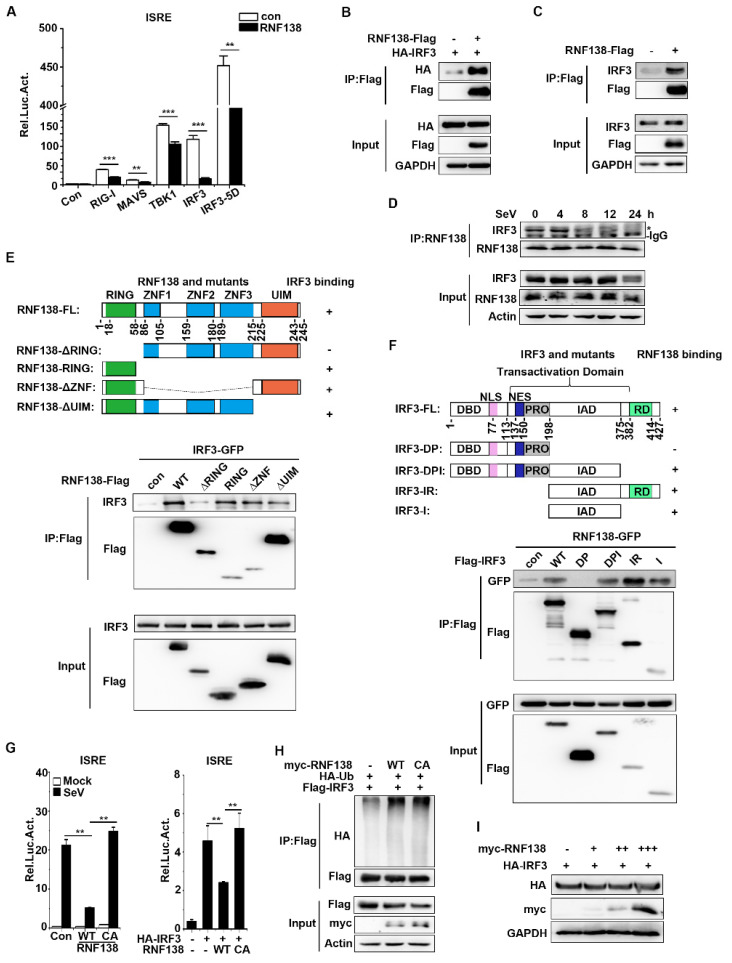
RNF138-mediated virus-triggered signaling at the level of IRF3. (**A**) The overexpression of RNF138 inhibited RIGI-, MAVS-, TBK1-, IRF3-, and IRF3-5D-mediated signaling. HEK293T cells were transfected with RNF138 or control plasmids together with the ISRE reporter and the indicated plasmids. Luciferase assays were performed 24 h after transfection. (**B**) Co-immunoprecipitation analysis of the interaction between RNF138 and IRF3 in HEK293T cells transfected with HA-IRF3 and RNF138-Flag or the control vector for 24 h. (**C**) Co-immunoprecipitation analysis of the interaction between RNF138 and IRF3 in HEK293T cells transfected with RNF138-Flag or control vector for 24 h. (**D**) Endogenous immunoprecipitation analysis of the interaction between RNF138 and IRF3 in HEK293T cells infected with SeV for the indicated times. * indicates the location of IRF3. (**E**,**F**) Domain mapping of the interaction of RNF138 with IRF3. HEK293T cells were transfected with the indicated truncations before co-immunoprecipitation and immunoblotting analysis with the indicated antibodies. The schematic presentations of the RNF138 and IRF3 truncations are shown at the top. (**G**) The effects of RNF138-WT and RNF138-CA (C18A/C54A) on the SeV-induced or IRF3-mediated activation of ISRE. HEK293T cells were transfected with the ISRE reporter and a control, RNF138-WT and RNF138-CA plasmids for 24 h, and then infected with SeV for 12 h (left), or transfected with the ISRE reporter and HA-IRF3 together with a control, RNF138-WT and RNF138-CA plasmids for 24 h (right panel) before luciferase assays. (**H**) The effects of RNF138 and its CA mutants on the polyubiquitination of IRF3. HEK293T cells were transfected with Flag-IRF3 and HA-Ub together with the control, RNF138-WT and RNF138-CA plasmids for 24 h, followed by immunoblotting and co-immunoprecipitation analysis with the indicated antibodies. (**I**) Immunoblot analysis at the protein level of IRF3 in HEK293T cells transfected with HA-IRF3 in fixed amounts and myc-RNF138 in different dosages. ** *p* < 0.01, *** *p* < 0.001 (unpaired *t*-test). Data represent at least two experiments with similar results (mean ± SD, n = 3 independent samples in (**A**–**C**,**G**)).

**Figure 5 ijms-24-16110-f005:**
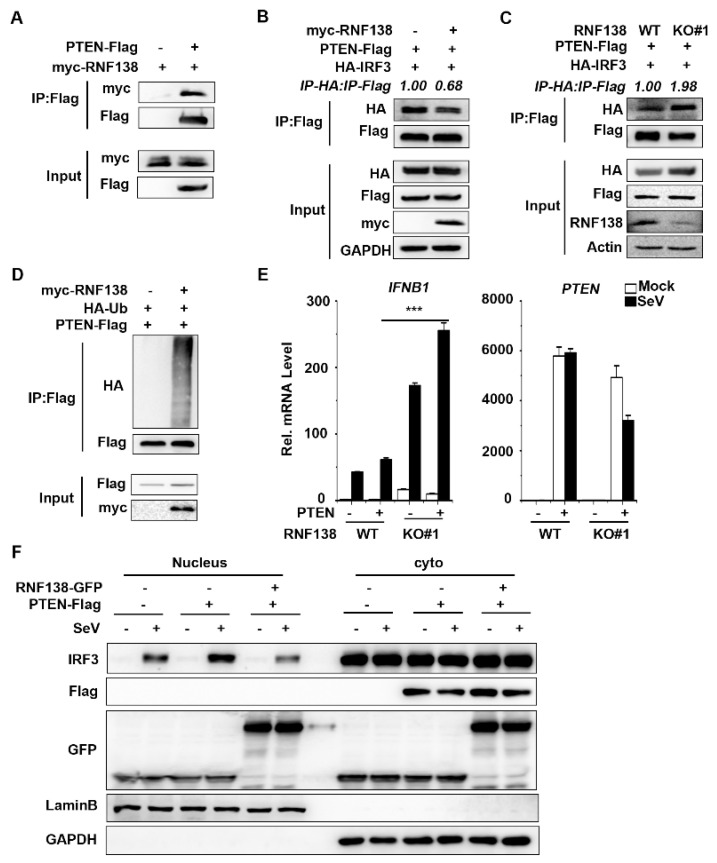
RNF138 inhibits IRF3 activation by ubiquitinating PTEN. (**A**) Co-immunoprecipitation analysis of the interaction between RNF138 and PTEN in HEK293T cells transfected with myc-RNF138 and PTEN-Flag or a control vector for 24 h. (**B**) RNF138 inhibited the association of PTEN with IRF3. HEK293T cells were transfected with PTEN-Flag and HA-IRF3 with a control and myc-RNF138 plasmid for 24 h before immunoblotting and co-immunoprecipitation analysis with the indicated antibodies. (**C**) RNF138 deficiency promotes the association of PTEN with IRF3. RNF138-KO and control HEK293T cells were transfected with PTEN-Flag and HA-IRF3 for 24 h before immunoblotting and co-immunoprecipitation analysis with the indicated antibodies. (**D**) The effects of RNF138 on the polyubiquitination of PTEN. HEK293T cells were transfected with PTEN-Flag and HA-Ub together with control or myc-RNF138 plasmid for 24 h, followed by immunoblotting and co-immunoprecipitation analysis with the indicated antibodies. (**E**) The effects of RNF138 deficiency on the PTEN-mediated transcription of IFNB1 gene. RNF138-KO and control HEK293T cells were transfected with PTEN-Flag for 24 h and then infected with SeV for 8 h before qPCR analysis. (**F**) The effects of RNF138 on the PTEN-mediated nuclear translocation of IRF3. HEK293T cells were transfected with the control or PTEN-Flag plasmid together with GFP or RNF138-GFP plasmid for 24 h then infected with SeV for 8 h. Immunoblot analysis of IRF3 in cytoplasmic (Cyto) and nucleus fractions with the indicated antibodies. *** *p* < 0.001 (unpaired *t*-test). Data represent at least two experiments with similar results (mean ± SD, n = 3 independent samples in (**A**,**D**,**E**)).

**Figure 6 ijms-24-16110-f006:**
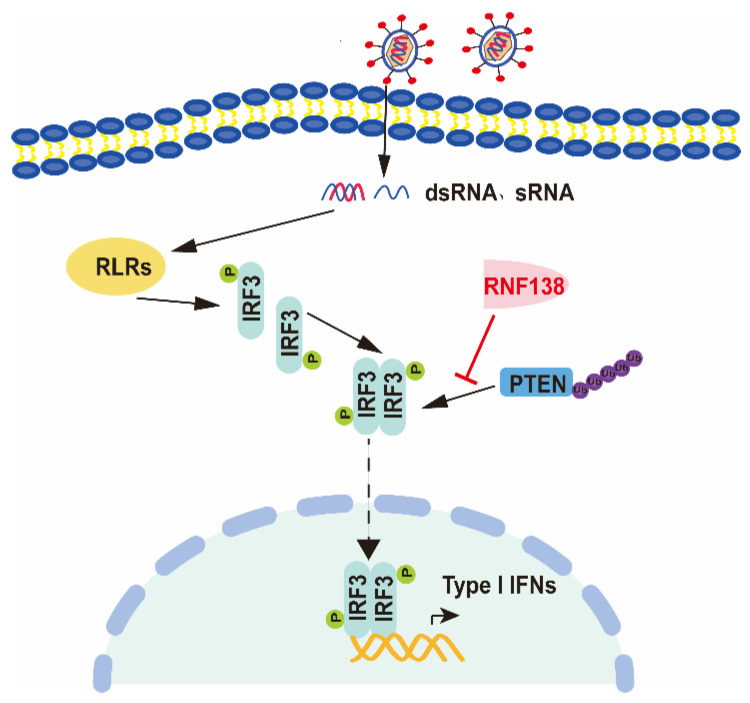
The proposed working model of RNF138 in the regulation of antiviral innate immune responses. RNF138 affects the ubiquitination of PTEN, which was ubiquitinated, likely inhibiting the interaction of IRF3 with PTEN and consequently inhibiting the import of IRF3 into the nucleus.

## Data Availability

All data are provided in the article and its Appendix A, or by the corresponding author upon reasonable request.

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
