# Peer review of "RNF138 Downregulates Antiviral Innate Immunity by Inhibiting IRF3 Activation"

_ijms, 2023, doi:10.3390/ijms242216110_

Round 1
Reviewer 1 Report
Comments and Suggestions for Authors
Review of Manuscript “RNF138 downregulates antiviral innate immunity by inhibiting IRF3 activation” by Xianhuang Zeng et al..
With the help of overexpression and knockout experiments the authors identify the E3 ubiquitin ligase RNF138 as a negative modulator of induction of type I interferon expression through the transcription factor IRF3. Although a direct interaction between RNF138 and IRF3 is demonstrated and the RING finger of RNF138 is required for this interaction, RNF138 does not seem to be directly involved in the polyubiquitination of IRF3 or lead to its degradation. Instead, RNF138 seems to act as an Ubiquitin E3 Ligase for PTEN, which previously had been described to release the negatively regulated phosphorylation of IRF3 at the serine residue 97 upon viral infection. These findings are quite interesting with regard to the induction of the innate immune response after induction of viral RNA’s by RIG-like receptors and in general the experiments seem to be well performed. However, many of the effects, especially those regarding the phosphorylation of IRF3 at serine residues 386 and 396 and its nuclear translocation, are rather marginal and their physiological relevance is therefore not quite clear. These concerns are listed in detail below. Furthermore, a revised version of the manuscript should improve the language in some passages, partly listed below under minor points.
Major points:
1) In the introduction section the authors focus exclusively on the mechanisms of type I interferon induction through the recognition of viral RNA’s in the cytoplasm by RIG-like receptors. Whereas Sendai Virus as a pathogen may be mainly recognized by such receptors in the cytoplasm, the authors should at least shortly describe the parallel pathway of recognition of viral RNA’s in the endosome by TOLL-like receptors (TLR), which also leads to type I interferon induction.
2) The reporter gene (luciferase) assays for determination of genes involved in interferon-induced immune response should be explained in more detail either in the Materials and Methods or the Results section
3) Fig. 1 C and 1D: The effects of RNF138-GFP overexpression on IRF3 phosphorylation at serine residues 386 and 396 (Fig. 1C) and its translocation to the nucleus seem to be rather marginal. Furthermore, in view of the inhibitory effects of GFP expression described, the appropriate control for the RNF138-GFP fusion protein would be GFP expression alone and not just the empty expression plasmid.
4) The immunoblot shown in the lower panel of fig. 2A is of rather low quality (high background smear).
5) Fig. 3A: The effect of the RNF138 knockout on IRF3-S396 phosphorylation in Sendai Virus infected RAW264.7 cells seem to be rather marginal.
6) Fig. 5B and 5C and corresponding text: no real effect of RNF138 overexpression or knockout on HA-IRF3 co-precipitation by PTEN-Flag can be observed (most upper panels).
7) Stimulation of IRF3 nuclear translocation by PTEN overexpression and inhibition of this effect by co-expression of RNF138-GFP are rather marginal.
Minor points:
Results section:
Section 2.1:
1) The first three to four sentences of the results section are not logical in their sentence structure (e.g. punctuation in the wrong place). Please rephrase.
2) Please explain abbreviation SeV (Sendai Virus) upon first appearance in the results section
Section 2.2:
1) First sentence, “to explored“, change to “to explore“.
2) First two sentences in the second section of the text: Please revise sentence structure (verb missing in second part of first sentence).
Section 2.3:
1) Please rephrase first sentence: instead of “is required“ please write “acts as“.
2) Second sentence in the third section of the text: Please revise logical sentence structure.
3) Fig. 4G and corresponding text: RNF138 enzyme inactivation mutant (CA) should be described in more detail.
Section 2.4:
1) Section describing experiments shown in fig. 5D: Please revise logical sentence structure.
2) The figure legends 5 D to F are in a wrong order!
3) Figure 5F: The placement of the labeling for SeV infection is not correct.
Comments on the Quality of English Language
English language should be improved in many passages.
Author Response
Thank you very much for your comments and professional advice.

Reviewer 2 Report
Comments and Suggestions for Authors
The research of Zeng et al. identified the E3 ubiquitin ligase RNF138 as an important negative regulator of virus-triggered IRF3 activation and IFN-β induction, and these results might be of great interest for innate immunologists. However, my main concern about Zeng et al. conclusions is that the Authors generalized their results as "antiviral", while they used only one virus model, which is, in addition, purely desribed in the text. It os unknow which strain of sendai virus was used, what was TCID50 or MOI, what were infection conditions. What is more, to prove "antiviral" association of the results, other viral models must be added.
Comments on the Quality of English LanguageThorough proofreading of the text is required as there are several English mistakes.
Author Response

(The authors gave the same response as above.)

Round 2
Reviewer 1 Report
Comments and Suggestions for Authors
Review of revised version (V2) of manuscript “RNF138 downregulates antiviral innate immunity by inhibiting IRF3 activation” by Xianhuang Zeng et al..
In the revised version of the manuscript and their point-by-point responses to my comments, the authors address most of the issues raised in my review of the original manuscript in an adequate fashion.
Only regarding my major point 1 adressing the neglect of the role of toll like receptors in the induction of the effector mechanisms of the innate immune system in the introduction section, I would have expected a little more information on the different subcellular locations of the RIG-like receptors and the TOLL-like receptors (please also mention at least all TLRs involved in the recognition of viral RNA’s) and the functional consequences.
Furthermore, the authors tend to generalize the findings made with only a single virus (Sendai Virus) as experimental system too far in the discussion often using the term of antiviral innate immune response. Such a generalization would require confirming the results after infection with other RNA viruses.
There is also still some room for improvements in the use of the English language. So it may be advisable to check this issue again.
Comments on the Quality of English LanguageThere is also still some room for improvements in the use of the English language. So it may be advisable to check this issue again.
Author Response
Dear reviewer,
Thank you very much for your comments and professional advice. These opinions help to improve academic rigor of our article. Based on your suggestion and request, we have made corrected modifications on the revised manuscript.

Reviewer 2 Report
Comments and Suggestions for Authors
I am satisfied with the current version of the manuscript.
Comments on the Quality of English LanguageMinor English mistakes, I would suggest thorough re-reading of the text.
Author Response
Dear reviewer,
Thank you very much for your comments and professional advice. These comments help to improve academic rigor of our article. Based on your suggestion and request, we have made corrected modifications on the revised manuscript.
